# Privatization of Biofilm Matrix in Structurally Heterogeneous Biofilms

Simon B. Otto,[a] Marivic Martin,[a,b] Daniel Schäfer,[b] Raimo Hartmann,[c] ⓘ Knut Drescher,[c,d] ⓘ Susanne Brix,[e] Anna Dragoš,[a] ⓘ Ákos T. Kovács[a,b]

[a]Bacterial Interactions and Evolution Group, Department of Biotechnology and Biomedicine, Technical University of Denmark, Kongens Lyngby, Denmark
[b]Terrestrial Biofilms Group, Institute of Microbiology, Friedrich Schiller University Jena, Jena, Germany
[c]Max Planck Institute for Terrestrial Microbiology, Marburg, Germany
[d]Department of Physics, Philipps-Universität Marburg, Marburg, Germany
[e]Disease Systems Immunology Group, Department of Biotechnology and Biomedicine, Technical University of Denmark, Kongens Lyngby, Denmark

Simon B. Otto and Marivic Martin contributed equally to this work. Author order was determined based on contribution to writing of the manuscript.

**ABSTRACT** The self-produced biofilm provides beneficial protection for the enclosed cells, but the costly production of matrix components makes producer cells susceptible to cheating by nonproducing individuals. Despite detrimental effects of nonproducers, biofilms can be heterogeneous, with isogenic nonproducers being a natural consequence of phenotypic differentiation processes. For instance, in *Bacillus subtilis* biofilm cells differ in production of the two major matrix components, the amyloid fiber protein TasA and exopolysaccharides (EPS), demonstrating different expression levels of corresponding matrix genes. This raises questions regarding matrix gene expression dynamics during biofilm development and the impact of phenotypic nonproducers on biofilm robustness. Here, we show that biofilms are structurally heterogeneous and can be separated into strongly and weakly associated clusters. We reveal that spatiotemporal changes in structural heterogeneity correlate with matrix gene expression, with TasA playing a key role in biofilm integrity and timing of development. We show that the matrix remains partially privatized by the producer subpopulation, where cells tightly stick together even when exposed to shear stress. Our results support previous findings on the existence of "weak points" in seemingly robust biofilms as well as on the key role of linkage proteins in biofilm formation. Furthermore, we provide a starting point for investigating the privatization of common goods within isogenic populations.

**IMPORTANCE** Biofilms are communities of bacteria protected by a self-produced extracellular matrix. The detrimental effects of nonproducing individuals on biofilm development raise questions about the dynamics between community members, especially when isogenic nonproducers exist within wild-type populations. We asked ourselves whether phenotypic nonproducers impact biofilm robustness, and where and when this heterogeneity of matrix gene expression occurs. Based on our results, we propose that the matrix remains partly privatized by the producing subpopulation, since producing cells stick together when exposed to shear stress. The important role of linkage proteins in robustness and development of the structurally heterogeneous biofilm provides an entry into studying the privatization of common goods within isogenic populations.

**KEYWORDS** *Bacillus subtilis*, biofilm, phenotypic heterogeneity, structural heterogeneity, exopolysaccharide

Biofilms are communities of tightly associated microorganisms encased in a self-produced extracellular matrix (1). This matrix provides shielding against biotic factors, such as antibiotics (2, 3) or natural competitors or predators (4, 5), and abiotic

**Ad Hoc Peer Reviewer** ⓘ Yunrong Chai, Northeastern University

Address correspondence to Anna Dragoš, adragos@dtu.dk, or Ákos T. Kovács, atkovacs@dtu.dk.

factors, such as harsh physicochemical (6) or shear (7) stress. As components of the biofilm matrix are costly to produce and they can be shared within the population, matrix producers are potentially susceptible to social cheating, where nonproducing mutants benefit from productive community members (8–10). This "tragedy of the commons" principle, in which nonparticipating users cannot be excluded from the use of common goods (9, 11, 12), has already been demonstrated for *Pseudomonas fluorescens* SBW25, for which exploitation by an evolved nonproducer resulted in biofilm collapse (13). Alternatively, production of the matrix components may not be easily exploitable if there is limited sharing, low cost of production, or spatial assortment of cells within the biofilm (14, 15). Finally, long-term cheating on matrix production may have evolutionary consequences for the producers, changing the phenotypic heterogeneity pattern of matrix expression within the population (16).

Although so-called "cheating" is traditionally associated with loss-of-function mutation in matrix genes, phenotypic nonproducers (cells in the so-called OFF state) can be an intrinsic part of clonal wild-type (WT) populations (17–19). For instance, in *Bacillus subtilis* NCBI 3610, a member of a probiotic and plant growth-promoting species (20, 21), the aforementioned phenotypic heterogeneity is fundamental to biofilm development, with individual cells exhibiting different tendencies to differentiate or express motility determinants (22, 23). Formation of pellicle biofilms, also referred to as "liquid-air interface" biofilms, in *B. subtilis* includes aerotaxis-driven swimming toward the liquid-air interface, subsequent motility loss, and adherent extracellular matrix production (24, 25). This differentiation of motile cells, becoming matrix-producing cells and spores, is not terminal, with genetically identical cells being able to alter their gene expression (26). While exploitability of the extracellular matrix by nonproducing mutants has been extensively studied, social interactions between clonal matrix producers and nonproducers in biofilms have been explored less. According to Hamilton's rule, altruistic sharing of public goods can easily evolve within isogenic populations, by means of inclusive fitness benefits (8). In other words, as long as the recipient carries the cooperative gene, cooperation should be evolutionarily stable in the absence of additional stabilizing mechanisms (27–30).

Still, it is not clear to what extent the matrix is shared between phenotypically heterogeneous producers and nonproducers, whether the presence of a nonproducing subpopulation has consequences for local biofilm robustness, and if/how the distribution of nonproducers changes during biofilm development. In fact, biofilms are nonuniform structures with variable local cell and polymer densities (31), which could be linked to different behavior of cells within a clonal population. Understanding how the heterogeneity of gene expression is linked to both biofilm development and structural robustness would provide better insight into the dynamics of biofilm communities.

The extracellular matrix of *B. subtilis* NCBI 3610 consists of two major components: an amyloid protein, TasA, and exopolysaccharide (EPS) (32). The EPS component is synthesized by protein products encoded by the *epsA-epsO* operon, with Δ*eps* mutants producing a weak and fragile biofilm (32, 33). The protein component TasA forms amyloid fiber-like structures (34, 35), and it is encoded in the *tapA-sipW-tasA* operon, with Δ*tasA* mutants unable to produce a biofilm (36). Mutant strains of *B. subtilis* NCBI 3610 lacking both operons cannot form a biofilm, whereas strains producing one of the components can complement each other and produce a wild-type-like pellicle (15, 32, 37). In this study, we demonstrate that, under exposure to shear stress, these seemingly sturdy pellicle biofilms disintegrate into extremely robust aggregates and single cells. We reveal that spatial and temporal changes in biofilm structural heterogeneity correlate with changes in expression of biofilm components, as cells in the ON state dominate within unbreakable biofilm aggregates. Therefore, despite inclusive fitness benefits from sharing the public goods within an isogenic population of producers, phenotypic cooperators (ON cells) still partially privatize the biofilm matrix. Further, we propose that the protein matrix component TasA plays a key role in maintaining biofilm robustness, with major consequences for the timing of development and the overall productivity of the biofilm. In general, our study links changes of phenotypic hetero-

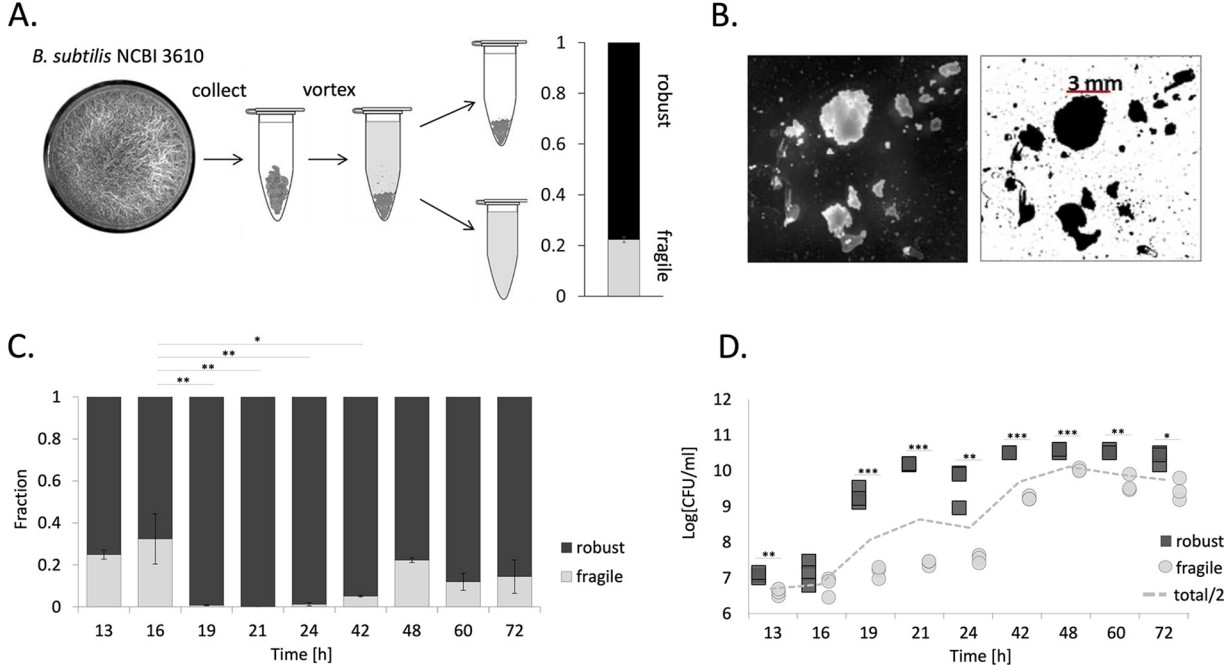

**FIG 1** Structural heterogeneity in pellicle biofilms. (A) Mechanical disruption of biofilms into "robust" and "fragile" fractions by vortexing the pellicle with sterile glass beads. The *y* axis of the graph indicates fraction of cells. (B) Microscopy images show that the robust fraction consists of nondispersible clumps that could be observed under a microscope with low magnification. These clumps were also present in 8-day-old pellicles. Bars, 5 mm. (C) Temporal changes in relative abundance of cells belonging to the robust and fragile fractions of the biofilm. The dark gray bar represents the robust fraction, while the light gray bar represents the fragile fraction. Data represent an average from biological triplicates, and error bars correspond to standard errors. (D) Changes in total cell counts in biofilm (dashed gray line) and cells in robust and fragile fractions over time were represented in logarithmic units. All data points were shown on the graph. For panels C and D, * stands for $P < 0.05$, ** stands for $P < 0.01$, and *** stands for $P < 0.001$.

geneity pattern with different stages of biofilm formation and reveals a fingerprint of such heterogeneity in biofilm structural robustness. It also reveals that privatization of public goods occurs even in isogenic microbial populations.

## RESULTS

**Structural heterogeneity develops in late stages of pellicle growth.** We began from a simple question: does phenotypic heterogeneity of matrix gene expression in *B. subtilis* (16, 19, 37) translate into nonuniform robustness within the biofilm? We will refer to such nonuniform biofilm robustness as "structural heterogeneity." Biofilms were mechanically disrupted by vortexing with sterile glass sand (Fig. 1A). Consequently, biofilm cells could be separated into two fractions: a fragile dispersible fraction and a robust nondispersible fraction, of which "clumps" could be easily observed under a microscope with low magnification and persisted for up to 8-day-old pellicles (Fig. 1B). This structural heterogeneity of the biofilm was predominantly observed in mature pellicles (older than 24 h), and the fractions were dynamic as the pellicle aged (Fig. 1C). During the establishment of a pellicle, around 13 to 16 h after inoculation of the bacteria into MSgg medium (defined in Materials and Methods), the roughly 0.25 to 0.32 fraction of cells could be assigned to the fragile fraction. Between 19 and 24 h, the juvenile pellicle was mostly structurally homogenous, as it consisted solely of a robust fraction. In later stages of pellicle development, the biofilm was again structurally heterogeneous, with an increase in cell number in the fragile fraction (Fig. 1C). To better understand the interplay between the robust and fragile fractions of biofilm during its development, we looked into changes in absolute cell numbers in both fractions (Fig. 1D). At most stages of biofilm development, the amounts of cells in robust and fragile fractions differed significantly (Fig. 1D). Moreover, biofilm development coupled with an increase of biofilm biomass could be divided into 2 stages: (i) early biofilm

development (between 16 and 21 h), which was characterized by a dramatic increase in total number of cells in the robust fraction (and only moderate increase of cells in the fragile fraction), and (ii) late biofilm development (between 24 and 48 h), when the number of cells in the robust fraction remained constant and cells in the fragile fraction largely increased in numbers (Fig. 1D). Overall, these results indicate that structural heterogeneity in mature pellicles (older than 24 h) results from the emergence of a loosely attached (fragile) population of cells on the "backbone" of robust cells. The turning point between early and late biofilm development takes place after around 24 h.

**Temporal changes in structural heterogeneity overlap changes in phenotypic heterogeneity.** Having shown that structural heterogeneity changes throughout biofilm development, we next sought to determine what underlies these changes. Considering that biofilms are nonuniform structures with variable polymer densities, we chose to investigate the expression of the *epsA-epsO* and *tapA-sipW-tasA* operons encoding the two major components of the biofilm, EPS and amyloid protein TasA, respectively (31, 32). Transcription levels were analyzed by flow cytometry using $P_{eps}$-*gfp* and $P_{tapA}$-*gfp* reporter strains at various pellicle ages (Fig. 2). Expression of both $P_{eps}$-*gfp* and $P_{tapA}$-*gfp* was shown to be low at 12 h (in most replicates insignificantly different from control, nonlabeled strain), when pellicles first emerged, indicating that most cells produced no or very small amounts of EPS and TasA. In both strains ($P_{eps}$-*gfp* and $P_{tapA}$-*gfp*), one replicate showed an emergence of an ON subpopulation, indicating biological stochasticity at this very early time point (Fig. 2; see also Data Set S1 in the supplemental material). The relative size of the ON subpopulation increased significantly between 12 and 16 h in both $P_{eps}$-*gfp* and $P_{tapA}$-*gfp*, thus in the majority of cells, 61% in $P_{eps}$-*gfp* and 66% in $P_{tapA}$-*gfp* (Fig. 2 and Data Set S1). Further changes were observed between 16 and 20 h, as the mean *eps* expression intensity increased significantly, so that the OFF subpopulation became the low-*eps* subpopulation (fluorescent signal significantly increased above the control level), and the ON subpopulation shifted toward a higher expression level. At the same time, the expression pattern of *tasA* became unimodal—with the OFF subpopulation disappearing completely (Fig. 2 and Data Set S1). Between 20 and 24 h, the *eps*-expressing subpopulation further increased in size, while the opposite was observed for the *tasA* expression pattern, where the OFF subpopulation reappeared (Fig. 2 and Data Set S1). At later time points, the heterogeneity level in both $P_{eps}$-*gfp* and $P_{tapA}$-*gfp* increased once again, with more pronounced OFF subpopulations. In mature pellicles (older than 24 h), similarly to the onset of biofilm formation, OFF subpopulations were in the majority (Fig. 2 and Data Set S1).

We observed similar changes in phenotypic heterogeneity when $P_{eps}$-*gfp* strains were analyzed under a confocal microscope (Fig. S1). Expression of *epsA-epsO* was most prevalent from 19 to 24 h with OFF subpopulations being observed at earlier and later time points. As images derived from intact biofilms, which contain clusters of ON and OFF cells, the bimodality of *eps* expression manifested after overlay of data from several frames per time point (Fig. S1). Overall, changes observed in phenotypic heterogeneity of *epsA-epsO* and *tapA-sipW-tasA* expression correlated with the temporal changes that we observed in structural heterogeneity of the biofilm. The so-called "turning point" in biofilm development, where growth of the robust fraction stops and growth of the fragile fraction begins (Fig. 1D), overlaps with a switch of *tasA* expression from unimodal ON state to bimodality and increasing numbers of *eps*-expressing cells. The late stage of biofilm development, when the fragile fraction increases in numbers (Fig. 1D), overlaps with an increase in relative sizes of OFF subpopulations with respect to both *eps* and *tasA*. This coupling between temporal changes in biofilm structural heterogeneity and matrix gene expression led us to hypothesize a spatial correlation between ON cells and the nondispersible parts of the biofilm.

**Expression of *epsA-epsO* and *tapA-sipW-tasA* operons in robust and fragile fractions.** To investigate if robustness is spatially related to high levels of polysaccharide and amyloid protein production, pellicles established by $P_{eps}$-*gfp* and $P_{tapA}$-*gfp*

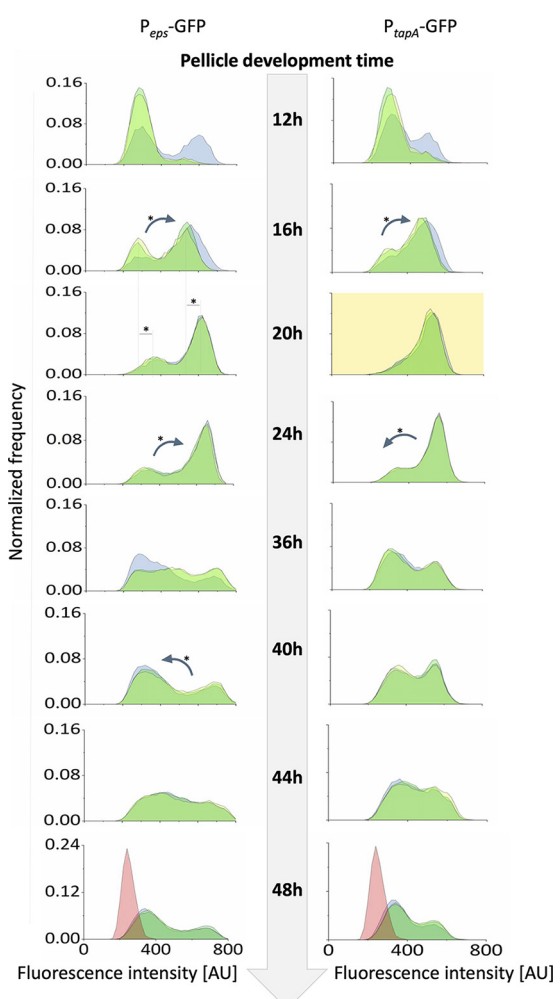

**FIG 2** Changes in matrix gene expression during biofilm development assessed by flow cytometry. Flow cytometry analysis showing distributions of fluorescence intensities of GFP-based transcriptional reporters for *epsA-epsO* (left) and *tapA-sipW-tasA* (right) at various time points throughout biofilm development. Histograms obtained for all biological replicates (n = 3) are overlaid for each time point. Data where distribution of matrix gene expression was unimodal (P$_{tapA}$-GFP, 20 h) are marked with a yellow background. Significant shifts of mean expression level in each subpopulation were indicated by dashed lines and asterisks. Significant changes in relative size of subpopulation with low- and high-matrix gene expression were shown as arrows (pointing toward shift direction) and asterisks. For changes in mean expression and subpopulation relative size, only significant differences between 2 neighboring time points were depicted on the image. Data for nonlabeled control were acquired for 48-h-old pellicle and integrated into the corresponding histograms as a red overlay. AU indicates arbitrary units.

reporter strains were mechanically disrupted, after which "clumps" and dispersible fractions were separately analyzed by flow cytometry (Fig. 3 and Data Set S1). Pellicles aged 24, 36, and 48 h were chosen because of the shift toward phenotypic heterogeneity that we had observed as the pellicles aged (Fig. 2 and Fig. S1). We noted that already at 24 h, there was a slight difference in *epsA-epsO* expression pattern between the robust and fragile pellicle fraction, as the percentage of ON cells was significantly larger in the robust fraction (Fig. 3A and Data Set S1). After 36 h, not only did the robust fraction of the P$_{eps}$-*gfp* strain contain a higher percentage of ON cells than did the fragile fraction, but also the *epsA-epsO* expression levels in the OFF subpopulation increased beyond the background noise, shifting the ON/OFF distribution toward a low-ON/high-ON scenario (Fig. 3A and Data Set S1). After 48 h, major changes took place in the robust fraction of P$_{eps}$-*gfp*, where the relative number of high-ON cells decreased and the low-ON subpopulation shifted back to an OFF state (Fig. 3A and Data Set S1). In contrast to P$_{eps}$-*gfp*, major differences between the robust and fragile

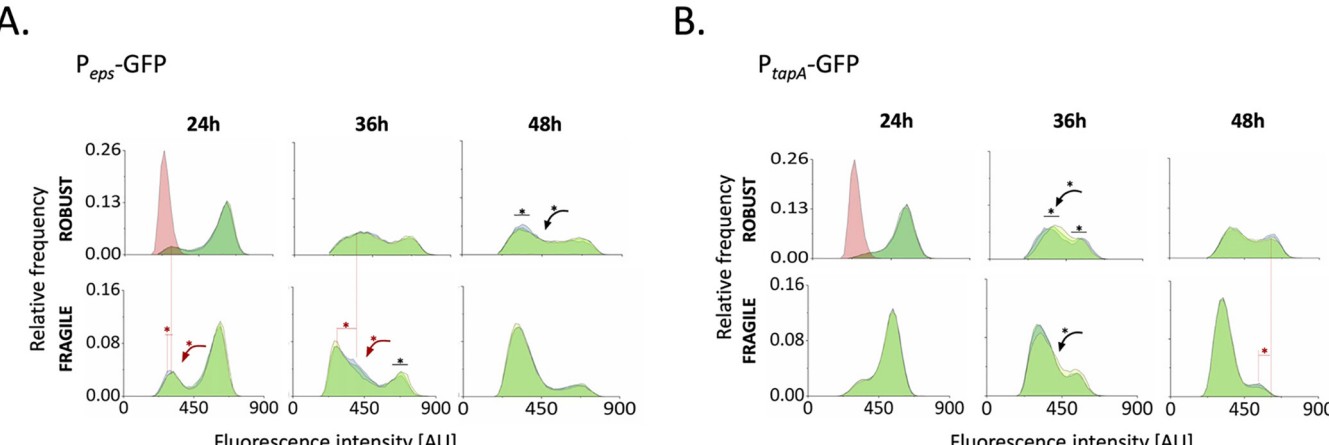

**FIG 3** Expression of matrix genes in robust and fragile fractions of the biofilm. Flow cytometry analysis showing average ($n = 3$) distributions of fluorescence intensities of mechanically disrupted $P_{eps}$-gfp (A) and $P_{tapA}$-gfp (B) reporter strains after 24, 36, and 48 h. The blue histogram represents the robust fraction, while the yellow graph represents the fragile fraction; the gray graph depicts nonlabeled cells. Data for nonlabeled control were acquired for 48-h-old pellicle and integrated into the upper left histograms of left and right panels, as a red overlay. AU indicates arbitrary units.

fractions of $P_{tapA}$-gfp were observed in late biofilms (after 48 h), where the robust fraction of the biofilm still contained a substantial amount of ON cells, with significantly higher expression levels than those observed in the fragile fraction of the biofilm (Fig. 3B and Data Set S1).

Overall, this analysis suggests that in early biofilm, fragile and robust fractions differ mostly in *eps* expression pattern (Fig. 3A and Data Set S1). On the other hand, in mature biofilms, when structural heterogeneity becomes more pronounced due to the increasing size of the fragile fraction (Fig. 1C and D), the robustness seems to be maintained through high levels of *tasA* expression (Fig. 3B and Data Set S1).

Additionally, we observed TasA nonproducers, cocultured with EPS nonproducers, to be dominant at the breakage points of clumps (Fig. S2), suggesting an involvement of TasA in biofilm integrity. Although our preliminary observation of increased abundance of the Δ*tasA* mutant at the pellicle breakage points requires further studies, it further points toward the importance of the TasA protein in biofilm mechanical robustness.

**TasA nonproducers have negative effects on the timing of pellicle development and final pellicle productivity.** Next, we aimed to determine how each matrix component affects biofilm development. First, we competed biofilm mutants lacking one or both matrix components against the wild type in competition assays with 1:1 relative inoculation. Relative fitness of biofilm mutants in the liquid medium was assessed after 24 and 48 h. Although Δ*eps* mutant and wild-type strains were equally fit in the pellicle, the mutant could outcompete the wild type in the liquid medium (below the pellicle biofilm) (Fig. 4A). In contrast, the Δ*tasA* mutant was clearly losing the competition against the wild type in the pellicle (Fig. 4A). Furthermore, the Δ*eps* Δ*tasA* mutant was significantly outcompeted in the pellicle and in the liquid after 48 h (Fig. 4A), which was clearly evident from the microscopy images of mixed pellicles (Fig. S3).

The reduced performance of the Δ*tasA* mutants in the pellicle suggests that it has negative effects on biofilm development. Thus, the effect of biofilm mutants on pellicle productivity (i.e., total number of cells in the pellicle) during development was assessed (Fig. 4B and Data Set S1). Cocultures of wild-type and biofilm mutants were mixed 1:1, and CFU productivity in the liquid and pellicle was determined at various time points throughout the development. We noted that both Δ*eps* and Δ*tasA* mutants significantly slowed down pellicle development, which was not the case for the Δ*eps* Δ*tasA* mutant (Fig. 4B and Data Set S1). The effect was especially pronounced for the Δ*tasA* mutant (Fig. 4B) and could also be captured by stereomicroscope time-lapse movies (Fig. S4

mSystems®

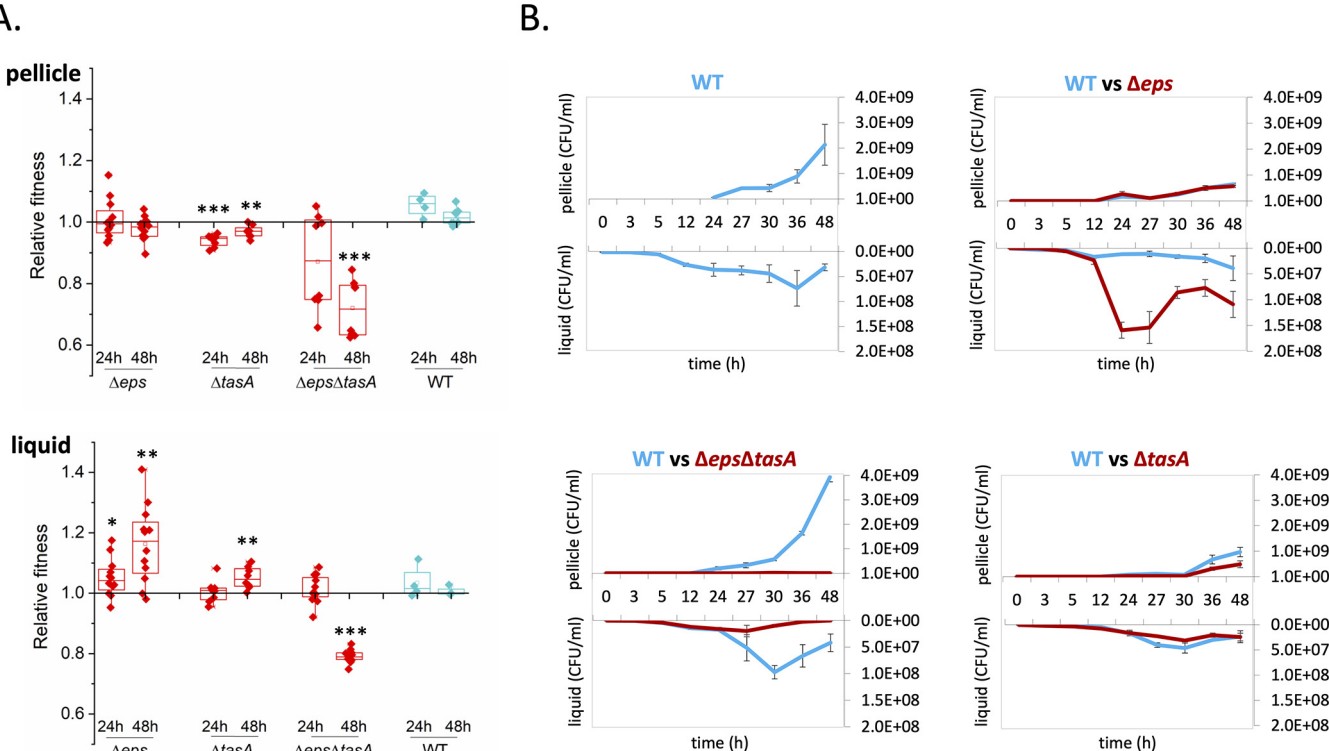

**FIG 4** Fitness of biofilm mutants in the pellicle and their effect on biofilm development. (A) Relative fitness of biofilm mutants in the pellicle biofilm (robust plus fragile fractions) and in liquid medium (below the biofilm) measured after 24 h and 48 h based on total CFU/ml counts. Boxes represent Q1 to Q3, lines represent the median, and bars span from maximum to minimum. * indicates $P < 0.05$; ** indicates $P < 0.01$; *** indicates $P < 0.001$. (B) Temporal changes in productivity during biofilm development in wild-type monoculture and in cocultures of the wild-type with either Δ*eps*, Δ*tasA*, or Δ*eps* Δ*tasA* strains. Productivity was assessed at different time points both in the pellicle and in liquid medium (below the biofilm). Pellicles were collected, resuspended in 1 ml of saline solution, disrupted, and serially diluted for CFU assay. CFU/ml stands for the number of cells obtained after pellicle disruption/1 ml of saline solution. Data points represent the average from $n = 3$ biological replicates, and error bars correspond to standard error.

and Movie S1). In conclusion, EPS, and especially TasA, nonproducers seem to slow down pellicle development and reduce final pellicle productivity (Fig. 4B, Fig. S4, and Movie S1). In addition, lack of a negative impact of the Δ*eps* Δ*tasA* mutant suggests that at least one of the two matrix components is required for positioning of the biofilm mutant in the pellicle and its negative effects on development and productivity.

**TasA nonproducers diminish pellicle robustness, while EPS nonproducers do not.** The function of TasA as a linkage protein and the importance of TasA for pellicle development suggest its significant contribution toward pellicle robustness. To investigate this, cocultures containing increasing percentages of Δ*eps* or Δ*tasA* mutant were mixed with the wild type and CFU productivity in the robust and fragile fractions of the pellicle was determined (Fig. 5). When the wild type was confounded with the Δ*eps* mutant, wild-type productivity in the robust fraction was reduced but its level was maintained as the proportion of the Δ*eps* mutant increased. Consistently, in both fragile and robust fractions we detected significant negative correlation between amounts of the WT and Δ*eps* strains (Pearson correlation = −0.85, $P < 1.2 \times 10^{-6}$; −0.61, $P < 0.004$; for fragile and robust fractions, respectively), suggesting that in both fractions, the mutant was able to compete with the WT (Fig. 5). Nevertheless, we did not detect significant correlation between the ratio of Δ*eps* mutant and biofilm robustness (Pearson correlation = 0.11, $P < 0.61$). In contrast, if the wild type was mixed with the Δ*tasA* mutant, it failed to incorporate into the robust pellicle fraction (Pearson correlation = 0.16, $P < 0.46$); however, it turned out to be detrimental for biofilm robustness (Pearson correlation = −0.85, $P < 1.6 \times 10^{-6}$). These results clearly show the negative impact of TasA nonproducers on pellicle robustness and the importance of TasA for incorporation into robust part of the biofilm.

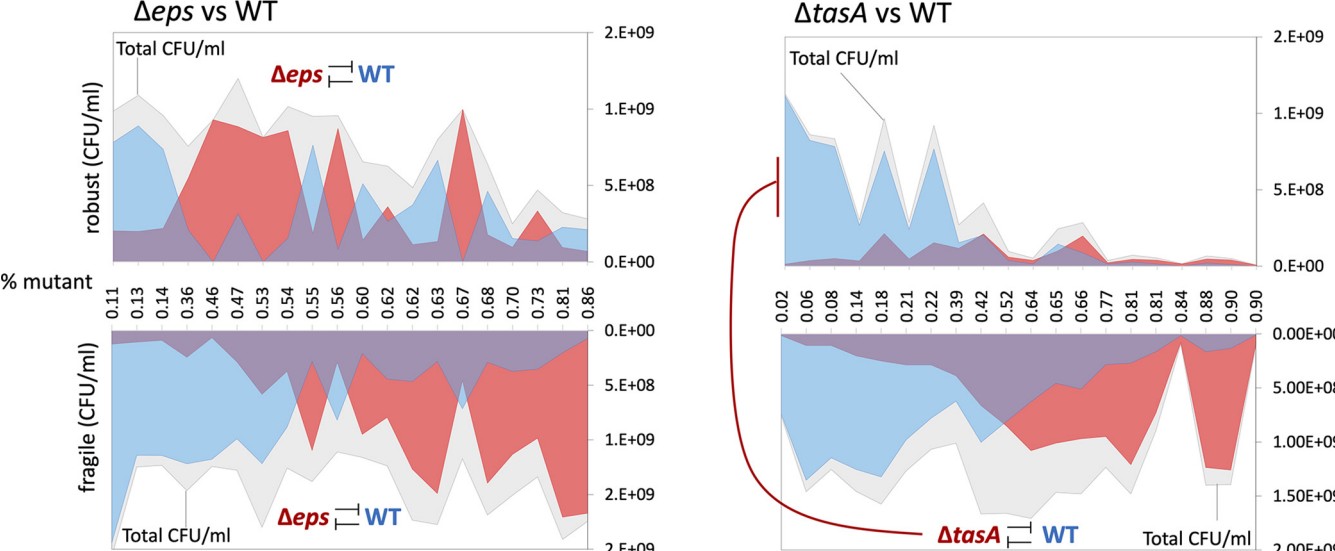

**FIG 5** Effect of biofilm mutants on pellicle robustness. Productivities of wild type and mutants based on total CFU/ml were assessed in mechanically disrupted robust and fragile fractions in cocultures of WT with increasing ratios of either Δeps or ΔtasA cells. Relationships between WT and mutants were examined using Pearson correlation coefficient. Significant negative correlations between WT and mutants, or between mutant and size of robust fraction, are labeled by inhibition symbol.

## DISCUSSION

Studies of the social interactions between genetically engineered matrix producers and nonproducers have become a common approach in sociobiology of biofilm communities (14, 37–39). Here, we addressed the consequences of native within-population phenotypic heterogeneity in matrix production for robustness, productivity, and timing of biofilm development. We revealed that production of matrix components shifts throughout biofilm development and that these changes correlate with temporal and spatial changes in biofilm robustness.

Biofilm development can be studied from different aspects (40–43). Here, we showed that in the initial stage of biofilm formation, the majority of the population was in an ON state, followed by heterogeneity in older biofilms. We show that in the time frame between 16 and 20 h, where an increase of robust pellicle biomass is the most pronounced (Fig. 1D), there is a significant shift in *eps* expression intensity and switching ON of *tasA* expression in nearly all biofilm cells. Therefore, our results link temporal dynamics in matrix gene expression with temporal changes of robust biofilm biomass.

These data are in line with previous studies, in which the spatiotemporal dynamics patterns of gene expression during the formation of submerged *Escherichia coli* biofilms were investigated (40). Moreover, bimodal expression of curli fibers was demonstrated, with high curli expression being confined to dense cell aggregates. The bimodal spatial expression of the structurally important curli fibers suggests a similar role for TasA, with these higher-cell-density aggregates providing protection against shear stress. Furthermore, in another study, the production of curli fibers was shown to protect the biofilm population against bacteriophage (41).

Importantly, cells exhibiting the ON state are more likely to occupy the most robust areas of the biofilm, thereby privatizing the benefit from matrix production under exposure to shear stress. Previous studies have shown that phenotypic heterogeneity of matrix production is present in biofilms of different species (17–19), with a similar phenomenon likely to occur in other biofilm-producing bacteria. Recently, quantitative visualization of *Pseudomonas aeruginosa* PAO1 aggregates has shown peak alginate gene expression in cells proximal to the surface compared with cells in the interior (44). Although it is likely that the interior of the *B. subtilis* pellicle biofilm contains more OFF

cells, we believe that the temporal shift that we observed from heterogeneity to homogeneity and then again toward heterogeneity is due to phenotypic differences between randomly distributed isogenic cells.

Accordingly, we revealed that TasA nonproducers have adverse effects on the timing of matrix development, productivity, and robustness, which was not the case for the EPS nonproducers. As EPS is likely costlier to produce and less privatized than TasA (37), the diminishing effects of Δ*tasA* may be linked to the specific structural role of TasA in the matrix, as could also be supported by its distinct localization pattern (45). The dominance of ON cells in the robust biofilm fraction was especially pronounced for *tasA* expression. Furthermore, we observed TasA nonproducers to be dominant at breakage points of biofilm clumps, suggesting these areas are weak points in biofilm integrity.

Conceivably, TasA functions similarly to the structural protein RmbA described in *Vibrio cholerae* biofilms, creating strong linkage between the producing cells (39). If the linkage role holds true, Δ*tasA* cells should be impaired in their ability to integrate into preestablished wild-type pellicles, which will be explored in the future. TasA was shown to have a strong adhesive role during interspecies interactions (46) and has been linked to structural integrity and physiology of *B. subtilis* biofilms (34, 47).

Our results support previous observations showing the importance of linkage proteins in formation of biofilms (32), as well as the presence of nonuniform biofilm structures (31). It remains to be discovered how the extracellular matrix remains privatized by ON cells and what are the ecological consequences or potential evolutionary benefits from biofilm structural heterogeneity. One possibility is bet-hedging, where weakly associated cells would adapt for short starvation periods, while early-sporulating aggregates are adapted for longer starvation periods, as proposed for slime molds (48). It remains to be tested whether robust and fragile fractions of *B. subtilis* biofilms differ in sporulation dynamics.

Our work has four major conclusions: (i) seemingly integral biofilms consist of robust and loosely associated cells, thereby being structurally heterogeneous; (ii) changes in the phenotypic heterogeneity pattern of matrix gene expression correlate with changes in biofilm structural heterogeneity in time and space; (iii) TasA nonproducers have detrimental effects on matrix development and structural integrity; and (iv) even in clonal microbial populations, where cooperation is stabilized by inclusive fitness benefits, public goods may be partially privatized by phenotypic producers.

## MATERIALS AND METHODS

**Bacterial strains and cultivation methods.** Strains used in this study are listed in Table 1. All strains were maintained in lysogeny broth (LB; LB-Lennox, Carl Roth), while MSgg medium (5 mM potassium phosphate buffer [pH 7], 100 mM morpholinepropanesulfonic acid [MOPS] [pH 7], 2 mM MgCl$_2$, 700 $\mu$M CaCl$_2$, 50 $\mu$M MnCl$_2$, 50 $\mu$M FeCl$_3$, 1 $\mu$M ZnCl$_2$, 2 $\mu$M thiamine, 0.5% [vol/vol] glycerol, 0.5% [wt/vol] glutamate) was used to induce biofilm formation (24). To obtain pellicle biofilms, bacteria were grown in static liquid MSgg medium at 30°C for 48 h, using 1% inoculum from overnight cultures. Prior to experiments, pellicles were sonicated according to an optimized protocol that allows for disruption of biofilms without affecting cell viability (15, 49). Productivity was determined by plating dilutions on LB agar to obtain CFU.

**Structural heterogeneity assay.** To assess structural heterogeneity of biofilms, pellicles were collected and transferred into a 1.5-ml microcentrifuge tube containing 1 ml of 0.9% (wt/vol) NaCl and ca. 20 $\mu$l of sterile glass sand ranging from 0.25 to 0.5 mm in grain size (Carl Roth). Next, pellicles were vortexed (Scientific Industries; Vortex-Genie 2) for 2 min at 3,200 rpm (maximal speed) and pellicle debris was allowed to sediment for 5 min. The dispersible fraction was transferred to a new Eppendorf tube, while the nondispersible "clumps" fraction was diluted in 1 ml of 0.9% (wt/vol) NaCl. Both fractions were sonicated as described previously (15), after which CFU levels were determined.

**Fitness assays.** To determine the fitness costs of EPS and TasA production, mKate2-labeled wild-type strains were competed with various biofilm-formation mutants. Overnight cultures were adjusted to the same optical density (OD) and mixed at a 1:1 ratio, and 1% coculture inoculum was transferred into 1.5 ml MSgg medium. Cocultures were grown under static conditions at 30°C. CFU levels in both sonicated pellicle and liquid medium were determined immediately after inoculation and after 24 or 48 h of growth. Wild-type colonies were distinguished from biofilm mutants based on pink color (visible emission from mKate2 reporter). The selection rate ($r$) was calculated as the difference in the Malthusian parameters of both strains: $r = \ln[\text{mutant } (t = 1)/\text{mutant } (t = 0)] - \{\ln[\text{wild type } (t = 1)/\text{wild type } (t = 0)]\}$, where $t = 1$ is the time point at which the pellicle was harvested (50).

**TABLE 1** Strains used in this study[a]

| Strain | Genotype | Reference |
|--------|----------|-----------|
| DK1042 | 3610 $comI^{Q12I}$ (wild type) | 54 |
| TB34 | DK1042 but $amyE$::P$_{hyperspank}$-$gfp$ (Cm$^r$) | 55 |
| TB35 | DK1042 but $amyE$::P$_{hyperspank}$-mKate2 (Cm$^r$) | 56 |
| TB500 | DK1042 but $amyE$::P$_{hyperspank}$-$gfp$ (Spec$^r$) | 55 |
| TB501 | DK1042 but $amyE$::P$_{hyperspank}$-mKate2 (Spec$^r$) | 37 |
| TB514 | DK1042 but $eps$::Tet$^r$, $tasA$::Spec$^r$, $amyE$::P$_{hyperspank}$-$gfp$ (Cm$^r$) | This study |
| TB515 | DK1042 but $eps$::Tet$^r$, $tasA$::Spec$^r$, $amyE$::P$_{hyperspank}$-mKate2 (Cm$^r$) | This study |
| TB524 | DK1042 but $eps$::Tet$^r$ , $amyE$::P$_{hyperspank}$-$gfp$ (Spec$^r$) | 37 |
| TB525 | DK1042 but $eps$::Tet$^r$ , $amyE$::P$_{hyperspank}$-mKate2 (Spec$^r$) | 37 |
| TB538 | DK1042 but $tasA$::Km$^r$, $amyE$::P$_{hyperspank}$-$gfp$ (Spec$^r$) | 37 |
| TB539 | DK1042 but $tasA$::Km$^r$, $amyE$::P$_{hyperspank}$-mKate2 (Spec$^r$) | 37 |
| TB601 | DK1042 but $eps$::Tet$^r$ | 49 |
| TB602 | DK1042 but $tasA$::Spec$^r$ | 55 |
| TB852 | DK1042 but $eps$::Tet$^r$, $tasA$::Km$^r$ | This study |
| TB863 | DK1042 but $tasA$::Km$^r$ | 37 |
| TB864 | DK1042 but $amyE$::P$_{hyperspank}$-mKate2 (Cm$^r$) $sacA$::P$_{eps}$-$gfp$ (Km$^r$) | 49 |
| TB865 | DK1042 but $amyE$::P$_{hyperspank}$-mKate2 (Cm$^r$) $sacA$::P$_{tapA}$-$gfp$ (Km$^r$) | 49 |

[a]TB852 was obtained by transforming DK1042 with genomic DNA isolated from TB601 and TB863 and selecting for tetracycline- and kanamycin-resistant colonies, respectively. Strains TB514 and TB515 were obtained by transforming TB34 and TB35, respectively, with genomic DNA obtained from TB601 and TB602 and selecting for tetracycline and spectinomycin resistance. Cm$^r$, Spec$^r$, Km$^r$, and Tet$^r$ denote chloramphenicol, spectinomycin, kanamycin, and tetracycline resistance cassettes, respectively.

**Flow cytometry.** Analysis was performed immediately after collection of the samples. To analyze expression levels of the $epsA$-$epsO$ and $tapA$-$sipW$-$tasA$ operons, flow cytometry analysis was performed using a BD FACSCanto II (BD Biosciences). One hundred thousand cells per sample were counted, where green fluorescent protein-positive (GFP$^+$) cells were detected by blue laser (488) via 530/30 and mKate2$^+$ cells were detected by red laser (633) and 660/20 filter, respectively. Three replicates per condition were incubated at 30°C for 12, 16, 20, 24, 36, 40, 44, or 48 h. Afterward, pellicles were harvested and sonicated. To study structural heterogeneity, harvested pellicles were vortexed as previously described, before sonication. Pellicles that were 12, 16, 20, or 24 h old were diluted 20 times, whereas pellicles that were 36, 40, 44, or 48 h old were diluted 200 times before flow cytometry analysis was performed. To obtain the average distribution of expression levels between replicates, data obtained from each replicate were subjected to binning using an identical bin size. Next, a mean count for each bin was obtained by averaging individual counts within this bin across all replicates, resulting in the mean distribution of single-cell-level expression per condition.

**Microscopy and image analysis.** To observe how biofilm mutants affect biofilm development, time-lapse microscopy experiments were performed. Overnight cultures were adjusted to the same optical density (OD), mixed in a 1:3 ratio (wild type to mutant), and inoculated in 500 $\mu$l MSgg medium inside an 8-well tissue culture chamber at 30°C (Sarstedt; width, 24 mm; length, 76 mm; growth area, 0.8 cm$^2$). Bright-field images of pellicles were taken with an Axio Zoom V16 stereomicroscope (×5 magnification; Carl Zeiss, Jena, Germany) equipped with a Zeiss CL 9000 LED light source, and an AxioCam MRm monochrome camera (Carl Zeiss), in which exposure time was set to 35 ms and images were captured every 15 min for a total of 48 h. Additionally, time-lapse videos of the wild-type monoculture biofilm development were recorded. For quantitative assessment of phenotypic heterogeneity, P$_{eps}$-$gfp$ pellicles were analyzed using a confocal laser scanning microscope (LSM 780; Carl Zeiss) equipped with a Plan-Apochromat/1.4 oil differential inference contrast (DIC) M27 63× objective and an argon laser (excitation at 488 nm for green fluorescence and 561 nm for red fluorescence, emission at 528 [±26] nm and 630 [±32] nm, respectively). Zen 2012 software (Carl Zeiss) and FIJI Image J software (51) were used for image recording and subsequent processing, respectively.

Confocal microscopy images were used to extract the single-cell-level distribution of $eps$-$gfp$ expression using our recently developed BiofilmQ software (52). This analysis involved the registration of image time series to avoid sample drift, followed by top-hat filtering to eliminate noise and Otsu thresholding to obtain a binary segmented image that separates the biofilm three-dimensional (3D) location from the background. The BiofilmQ-inbuilt technique was used for dissecting this 3D volume into pseudocell cubes, which have the same volume as an average *B. subtilis* cell (4.6 $\mu$m$^3$) (53), based on the mKate2 fluorescence (constitutively expressed in all cells). Next, we quantified the GFP signal per pseudocell and plotted its distribution at different time points.

**Statistical analysis.** Statistical differences between two experimental groups (e.g., total CFU/ml in robust biofilm fraction versus total CFU/ml in fragile biofilm fraction at a single time point) were assessed using a two-sample $t$ test assuming equal variance. One-way analysis of variance (ANOVA) and the Tukey test were used for multiple-sample comparisons (e.g., robust biofilm fraction across all sampling time points, where destructive sampling was applied). Two-way ANOVA and the Tukey test were used to assess the effects of time and different mutant on the wild type (WT) in pellicle and in the liquid fraction. Correlations were assessed using the Pearson correlation coefficient. No statistical methods were used to predetermine sample size, and the experiments were not randomized. All statistical tests were performed using OriginPro 2018 software.

## SUPPLEMENTAL MATERIAL

Supplemental material is available online only.

**FIG S1**, PDF file, 0.4 MB.
**FIG S2**, PDF file, 0.5 MB.
**FIG S3**, PDF file, 0.6 MB.
**FIG S4**, PDF file, 0.7 MB.
**DATA SET S1**, XLSX file, 0.1 MB.
**MOVIE S1**, MOV file, 2.3 MB.

## ACKNOWLEDGMENTS

This study was supported by the Deutsche Forschungsgemeinschaft (DFG) to Á.T.K. (KO4741/2.1) within the Priority Program SPP1617, and the Collaborative Research Center SFB987 (to K.D.). S.B.O. and M.M. were supported by an Erasmus+ fellowship and a FEMS Research and Training grant (FEMS-RG-2017-0054), respectively. This project has received funding from the European Union's Horizon 2020 research and innovation program under the Marie Skłodowska-Curie grant agreement no. 713683 (H.C. Ørsted COFUND to A.D.) and the European Research Council (StG-716734 to K.D.). Work in the laboratory of Á.T.K. is partly supported by the Danish National Research Foundation (DNRF137) for the Center for Microbial Secondary Metabolites.

Á.T.K. and A.D. conceived the project; S.B.O., M.M., D.S., R.H., and A.D. performed experiments; K.D. and S.B. contributed methodologies and equipment, respectively; S.B.O., A.D., and Á.T.K. wrote the manuscript, with all authors contributing to the final version.

We declare we have no competing interests.

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
