## [Reviewer comments · mSystems]

Privatization of biofilm matrix in structurally heterogeneous biofilms

Simon B. Otto, Marivic Martin, Daniel Schäfer, Raimo Hartmann, Knut Drescher, Susanne Brix, Anna Dragoš, and Ákos T Kovács

Corresponding Author(s): Ákos T Kovács, Technical University of Denmark

Review Timeline:

Submission Date:	May 11, 2020
Editorial Decision:	June 24, 2020
Revision Received:	July 8, 2020
Accepted:	July 13, 2020

Editor: Mark Mandel

Reviewer(s): Disclosure of reviewer identity is with reference to reviewer comments included in decision letter(s). The following individuals involved in review of your submission have agreed to reveal their identity: Yunrong Chai (Reviewer #1)

Transaction Report:

DOI: <https://doi.org/10.1128/mSystems.00425-20>

June 24, 2020

Prof. Ákos T Kovács
Technical University of Denmark
Department of Biotechnology and Biomedicine
Søltofts Plads 221
Kgs Lyngby 2800
Denmark

Re: mSystems00425-20 (Privatization of biofilm matrix in structurally heterogeneous biofilms)

Dear Prof. Ákos T Kovács:

I asked two experts in the field to review the manuscript that you and your coauthors submitted. As you can see below, both reviewers were impressed with the work and think that the study would be of interest to readers of mSystems. They both have questions about data interpretation, methods, and controls that need to be addressed before the manuscript can be accepted for publication. I also read the manuscript and found the study to be well conducted and a significant contribution. I hope that the thoughtful comments offered by the reviewers are helpful to improve the final manuscript.

To submit your modified manuscript, log onto the eJP submission site at <https://msystems.msubmit.net/cgi-bin/main.plex>. If you cannot remember your password, click the "Can't remember your password?" link and follow the instructions on the screen. Go to Author Tasks and click the appropriate manuscript title to begin the resubmission process. The information that you entered when you first submitted the paper will be displayed. Please update the information as necessary. Provide (1) point-by-point responses to the issues raised by the reviewers as file type "Response to Reviewers," not in your cover letter, and (2) a PDF file that indicates the changes from the original submission (by highlighting or underlining the changes) as file type "Marked Up Manuscript - For Review Only." This file should be computer-generated to show the changes between the two manuscripts (e.g., by Microsoft Word's "Compare Documents" function).

Due to the SARS-CoV-2 pandemic, our typical 60 day deadline for revisions will not be applied. I hope that you will be able to submit a revised manuscript soon, but want to reassure you that the journal will be flexible in terms of timing, particularly if experimental revisions are needed. When you are ready to resubmit, please know that our staff and Editors are working remotely and handling submissions without delay. If you do not wish to modify the manuscript and prefer to submit it to another journal, please notify me of your decision immediately so that the manuscript may be formally withdrawn from consideration by mSystems.

To avoid unnecessary delay in publication should your modified manuscript be accepted, it is important that all elements you upload meet the technical requirements for production. I strongly recommend that you check your digital images using the Rapid Inspector tool at <http://rapidinspector.cadmus.com/RapidInspector/zmw/>.

If your manuscript is accepted for publication, you will be contacted separately about payment

when the proofs are issued; please follow the instructions in that e-mail. Arrangements for payment must be made before your article is published. For a complete list of **Publication Fees**, including supplemental material costs, please visit our website.

Sincerely,

Mark Mandel

Editor, mSystems

Journals Department
Reviewer comments:

Reviewer #1 (Comments for the Author):

This study by Otto et al. demonstrated structural heterogeneity of the *Bacillus subtilis* pellicle biofilm and dynamics of such heterogeneity during biofilm development. The authors tried to suggest that matrix gene expression heterogeneity in individual cells may drive the structural heterogeneity. In the second part of the study, the authors tried to assess the impact of two different matrix components, EPS and TasaA, on pellicle biofilm robustness and dynamic development using genetic mutants, and measurement of biofilm robustness via cfu counting.

Overall, the discussion of bacterial sociology, and the social behavior question during *B. subtilis* development that the authors tried to address are fascinating. Some of the results presented in this study are very interesting. This manuscript is well written and joyful to read. I do have somewhat different opinions about the two parts of this study, 1) structural heterogeneity (Fig.1-3) and 2) the impact of two the matrix components on biofilm robustness and development (Fig. 4-5), especially the second part. In the second part of the study, based on mainly the competition assays between WT and the mutants, the authors concluded that TasaA is pivotal to biofilm robustness and development, while EPS is much less so, here I have hesitation whether the major conclusions are fully justified by data.

Major points:

Fig. 3, although the idea of testing the matrix gene expression in separated fractions of the pellicle (based on robustness) is quite interesting, how do the results here truly (or fully) support the idea

that matrix production of cells drive the separation of fragile and robust fractions of the pellicle (as indicated by the authors ?) The reason I argue is because the pattern of the matrix gene expression (both *eps* and *tasA*) seems not much different between the robust and the fragile fractions to support the clear phenotypic difference.

Lines 205-209, I tend to disagree with how the authors interpret the result shown in Fig. S2. The direct conclusion from Fig. S2, seems to me, is that EPS-producers out-compete (probably not dominate) the *TasA*-producers in a mixed pellicle. I wonder why this necessarily leads to the conclusion of higher importance of *TasA* (than EPS?) in biofilm integrity.

Lines 217-218, I wonder if this is well supported by fig. 4a, which shows 0.9-0.95 relative abundance of *tasA* vs WT (meaning 5-10% less than the WT) at 48 hr.

Fig. 4a and 4b, can the authors explain how is 4a different from 4b except that more time points were used in 4b ? second, some of the cfu data in 4a do not exactly match 4b, for example, the *tasA* relative abundance at 48 hours showing 0.9-0.95 (in pellicle) in 4a while in 4b, the relative abundance is more like 0.5 based on cfu reading (also in pellicle), are they different sets of experiments? can the authors explain?

Fig5, *eps* vs WT, based on CFU counting, one would predict phenotypically, the overall robustness of the biofilm would diminish when *eps* vs WT ratio increases, which would lead to the opposite conclusion that EPS-nonproducers have no impact the overall pellicle robustness. The same can be applied to *tasA* vs WT although to a more severe degree.

Line 237, I would suggest the authors also add the results of the overall pellicle robustness as a support of the subtitle.

Minor points:

Fig. 1C, why Y-axis has a scale up to 0.5, is it supposed to be 1.0 (100%) ?

Line 181, This title may be supported in the assays of the total pellicle, but not in the separated robust and fragile fractions, I think, based on

Line222, the word "exclusion" seems too strong for the same reason described above.

Reviewer #2 (Comments for the Author):

This manuscript by Otto et al examines the structural heterogeneity within pellicle biofilms formed by *Bacillus subtilis*, and the underlying matrix gene expression within robust and fragile segments of the biofilm. Overall this is an interesting, and well-done investigation, that reveals significant links between the expression of the biofilm matrix components and the localised structural stability of the biofilm. The experiments have all been well-designed, and robustly performed and analysed. There are some points at which additional information needs to be provided, and some minor text changes, but overall this is a well-written, easy-to-read manuscript.

Significant points:

Line 93: (and elsewhere) you talk about *TasA* providing structure/rigidity but you don't mention it forming fibre-like structures

Line 117: you refer to sterile glass sand here (elsewhere you call it glass beads, Line 545) - is there a typically particle size of the sand? It would be useful to have this information to judge the type/amount of forces being applied to the biofilm to separate the robust and fragile fractions.

Line 148 and Figure 2: you talk about the comparison to a non-gfp control - but you don't show this control. It needs to be added to the figures and/or added as a supplemental figure.

Line 156 and Figure 2: again you talk about the control level but don't show it, it needs to be added somewhere to show that the differences shown are true

Line 191 and Figure 3: you talk about background noise, but you do not show the background on the graphs. Was the non-fluorescent background pellicle only analysed once? Or was a separate pellicle analysed at each time point? Does the background levels change with time (ie does any autofluorescence appear/increase?) Could it be related to the presence of more EPS itself in the robust fraction?

Figure 2: The peak shifts in the eps readouts at 20hour are unusual - the background non gfp control is also needed to be shown. Also if the same shift occurred in the tapA readout (which the ON population looks like it has compared to 16hour) then there may be a small OFF peak in this sample

Figure 4B: Wt only sample: why is the T=0 liquid sample CFU value so high only in this one sample?

Figure 5: There is a lot of variation between the results with different starting ratios - whilst there are clear trends, there are some obvious outliers (eg 0.84 ratio sample in tasA vs WT). Can this be explained or is it simply an experimental anomaly?

Minor text changes:

Line 43: 'raise' should be 'raises'

Line 70: 'member of probiotic' should be 'member of a probiotic'

Line 73: 'Formation of pellicle biofilm' should either be 'Formation of a pellicle biofilms' or 'Formation of pellicle biofilms'

Line 74: 'towards liquid-air' should be 'towards the liquid-air'

Line 74: 'biofilm, in' should be 'biofilms, in'

Line 77: 'of extracellular' should be 'of the extracellular'

Line 88: 'variable and local' should be 'variable local'

Line 105: 'that protein' should be 'that the protein'

Line 119: 'magnifications' should be 'magnification'

Line 119: 'persisted up' should be 'persisted for up'

Line 121: can you add some timings to 'mature'?

Line 128: 'Most' should be 'At most'

Line 133: add hours after 24-48

Line 134: 'when number' should be 'when the number'

Line 134: 'and cells of fragile' should be 'and cell in the fragile'

Line 136: 'from emergence' should be 'from the emergence'

Line 152: 'of ON-' should be 'of the ON-'

Line 157: 'towards higher' should be 'towards a higher'

Line 158: 'with OFF-' should be 'with the OFF-'

Line 160: 'where OFF-' should be 'where the OFF-'

Line 161: 'points, heterogeneity' should be 'points, the heterogeneity'

Line 185: 'Pellicle with' should be 'pellicles of ages'

Line 192: 'towards low' should be 'towards a low'

Line 195: 'difference' should be 'differences'

Line 196: 'fragile' should be 'robust' based on the data?

Line 200: 'suggest' should be 'suggests'

Line 202: 'to increasing' should be 'to the increasing'

Line 203: 'seem' should be 'seems'

Line 207: you say observation of increased abundance of TasA - but you have not shown increased protein levels. You saw an increased number of delta-tasA cells?

Line 208: remove 'out'

Line 215: 'in both the' - you only say one sample, need to add in the pellicle as well?

Line 218: 'loosing' should be 'losing'

Line 220: 'that' should be 'which'

Lines 222-224: an extra line break here? This sentence is orphaned, and should be a part of the

following paragraph?

Line 239: 'suggest' should be 'suggests'

Line 250: 'up' should be 'out'

Line 298: 'how extracellular' should be 'how the extracellular'

Line 324: More info is needed here regarding vortexed - what settings were used? The force/speed of the vortex will have a big impact on the shear force applied

Line 349: Was the flow cytometry done immediately after sample collection or were the samples fixed and stored? Could this be the cause of the unusual changes in peak intensity at certain time points?

Line 381, 382 and 385, 556: 'access' or 'accessed' should be 'assess' or 'assessed'

Line 549: you talk about blue and yellow bars, but the figure is in greyscale

Table 1: There are strains used in the supplemental figures that are not included in the strains table, (eg the WT gfp strain), these should be added for completeness

Figure 1C: In the text you talk about the different populations in terms of %, but the yaxis here is as a fraction - it would be better to be consistent

Figure 2: The third repeat isn't visible in most of the graphs - could the colouring be changed to ensure all three repeats are visible?

Figure 4B: the volume that the pellicle was resuspended in should be given in the legend and/or the pellicle axes labelled simply as CFU/pellicle

Figure 5: It might be helpful to also include a curve of 'total cell number' to see how the total number of cells varies in each fraction of the biofilm with the different starting ratios, this may help with the anomalous samples like the 0.84 ratio in *tasA* vs WT

Fig S1 Legend: 'accessed' should be 'assessed'

Fig S1: were all the images taken at the same depth/height in the pellicle at all timepoints?

Fig S3: A scale bar is needed for these images. Why has this figure been false-coloured blue and yellow when all others are in green/red? Yellow and blue is preferential for colour blindness, but it would be best to be consistent.

Reviewer #1 (Comments for the Author):

This study by Otto et al. demonstrated structural heterogeneity of the *Bacillus subtilis* pellicle biofilm and dynamics of such heterogeneity during biofilm development. The authors tried to suggest that matrix gene expression heterogeneity in individual cells may drive the structural heterogeneity. In the second part of the study, the authors tried to assess the impact of two different matrix components, EPS and TasA, on pellicle biofilm robustness and dynamic development using genetic mutants, and measurement of biofilm robustness via cfu counting.

Overall, the discussion of bacterial sociology, and the social behavior question during *B. subtilis* development that the authors tried to address are fascinating. Some of the results presented in this study are very interesting. This manuscript is well written and joyful to read. I do have somewhat different opinions about the two parts of this study, 1) structural heterogeneity (Fig. 1-3) and 2) the impact of two the matrix components on biofilm robustness and development (Fig. 4-5), especially the second part. In the second part of the study, based on mainly the competition assays between WT and the mutants, the authors concluded that TasA is pivotal to biofilm robustness and development, while EPS is much less so, here I have hesitation whether the major conclusions are fully justified by data.

Major points:

Fig. 3, although the idea of testing the matrix gene expression in separated fractions of the pellicle (based on robustness) is quite interesting, how do the results here truly (or fully) support the idea that matrix production of cells drive the separation of fragile and robust fractions of the pellicle (as indicated by the authors ?) The reason I argue is because the pattern of the matrix gene expression (both *eps* and *tasA*) seems not much different between the robust and the fragile fractions to support the clear phenotypic difference.

We agree that one might expect more dramatic differences in matrix gene expression in robust and fragile fractions of pellicle biofilm. However, as for the first attempt to challenge the hypothesis on structural heterogeneity of pellicle biofilms, using rather rudimentary protocol to separate robust and fragile biofilm fractions, we argue it is fascinating that statistically significant differences do occur (Fig3).

With current 'crude' fraction separation approach and statistical power we can already state:

- *There are significantly more *eps*-OFF cells in fragile fraction compared to robust fraction after 24h and 36h*
- *The expression level of P_{tasA} is significantly higher in the robust fraction after 48h*
- *There are significant differences between how *tasA*- and *eps*-deficient strains influence biofilm robustness*

*We are convinced that optimization of fraction separation protocol in the future, will only emphasize these differences, and allow broader studies (e.g. transcriptomics) on role of structural heterogeneity in biofilms (beyond *B. subtilis* species).*

Lines 205-209, I tend to disagree with how the authors interpret the result shown in Fig. S2. The direct conclusion from Fig. S2, seems to me, is that EPS-producers out-compete (probably not dominate) the TasA-producers in a mixed pellicle. I wonder why this necessarily leads to the conclusion of higher importance of TasA (than EPS?) in biofilm integrity.

We understand the above concern and it was not our intention to lay out Fig S2 as an evidence for higher importance of TasA in biofilm robustness, which manifests also in our cautious comments of S2 in the results section. The data presented in Fig S2 confirms previous description on the distribution of strains in optimal productivity (as reported in Dragos et al 2018 Current Biology). Nevertheless, we did demonstrate (data in Fig 5), that increasing ratio of TasA-non-producers is negatively correlated with biofilm robustness – which is not the case for EPS-non-producer. Figure S2 is presented here as a complementary and (in our opinion) interesting observation that TasA-non-producers tend to reside at the biofilm breakage points. We think this result, although qualitative and not evidential, may still inspire biofilm researchers to take ‘out of the box’ approach while studying structural role of biofilm components in co-culture setup.

Lines 217-218, I wonder if this is well supported by fig. 4a, which shows 0.9-0.95 relative abundance of tasA vs WT (meaning 5-10% less than the WT) at 48 hr.

Figure 4A represents relative fitness of the mutant compared to the WT (and not relative abundance): value 0.9-0.95 indicates mutant is doing slightly worse compared to the WT. Perhaps our answer to below comment will also help to clarify this concern.

Fig. 4a and 4b, can the authors explain how is 4a different from 4b except that more time points were used in 4b ? second, some of the cfu data in 4a do not exactly match 4b, for example, the tasA relative abundance at 48 hours showing 0.9-0.95 (in pellicle) in 4a while in 4b, the relative abundance is more like 0.5 based on cfu reading (also in pellicle), are they different sets of experiments? can the authors explain?

Certain degree of variation between 4a and 4b is expected – data are coming from 2 independent experiment, that were in fact performed by different authors, and at different locations of our research lab (4a performed at DTU, Denmark, 4b performed at FSU Jena in Germany). The referee is right about the major difference in experimental design being finest temporal scale assessed in 4b. Figure 4a is showing relative fitness of each mutant (and WT as a control) in pellicle and in the liquid, while 4b is showing changes in absolute CFU of WT and mutants in pellicle and liquid fraction. Reg. data for tasA – fitness is below 1 in the pellicle (4a), what also manifest in lower absolute CFU compared to the WT in pellicle (4b). Similar stands for other mutants, with Δeps being equally fit in pellicle (4a, fitness around 1 – similar CFU at 4b), but significantly more fit in the liquid (see 4a, fitness value above 1, and CFU in the liquid higher compared to the WT).

Fig5, eps vs WT, based on CFU counting, one would predict phenotypically, the overall robustness of the biofilm would diminish when eps vs WT ratio increases, which would lead to the opposite conclusion that EPS-nonproducers have no impact the overall pellicle robustness. The same can be applied to tasA vs WT although to a more severe degree.

We appreciate the above comment, however we would like to defend our analysis on the level of significance that demonstrated the significant impact of $\Delta tasA$, but not Δeps on overall pellicle robustness: see the statistical analysis of data shown in Fig 5 – we were able to show significant negative correlation between ratio of $\Delta tasA$ in the pellicle and size of robust pellicle fraction (Pearson corr. = -0.85, $p < 1.6 \times 10^{-6}$), which is not the case for Δeps (Pearson corr. = 0.16, $p < 0.46$), showing more randomness in the relationship between ratio of the mutant and size of robust fraction.

Line 237, I would suggest the authors also add the results of the overall pellicle robustness as a support of the subtitle.

Figure 5 was updated including overall CFU/ml of robust fraction and fragile fraction – acc. to the above comment, as well as comment of Reviewer 2.

Minor points:

Fig. 1C, why Y-axis has a scale up to 0.5, is it supposed to be 1.0 (100%) ?

The scale was deliberately setup to 0.5, to better illustrate the two fractions – as fragile fraction is in the minority. We have adjusted Fig. 1A, which is now representing more accurate ratios between fragile and robust fraction. To accommodate the request of the Reviewer, we have also adjusted the scale of Fig 1C.

Line 181, This title may be supported in the assays of the total pellicle, but not in the separated robust and fragile fractions, I think, based on

We understand the above concern and therefore we had smoothed the paragraph title to remove the strong claim.

Line222, the word "exclusion" seems too strong for the same reason described above.

We agree, the word was replaced with reduced performance.

Reviewer #2 (Comments for the Author):

This manuscript by Otto et al examines the structural heterogeneity within pellicle biofilms formed by *Bacillus subtilis*, and the underlying matrix gene expression within robust and fragile segments of the biofilm. Overall this is an interesting, and well-done investigation, that reveals significant links between the expression of the biofilm matrix components and the localised structural stability of the biofilm. The experiments have all been well-designed, and robustly performed and analysed. There are some points at which additional information needs to be provided, and some minor text changes, but overall this is a well-written, easy-to-read manuscript.

Significant points:

Line 93: (and elsewhere) you talk about TasA providing structure/rigidity but you don't mention it forming fibre-like structures

Corrected, we include this information in the introduction.

Line 117: you refer to sterile glass sand here (elsewhere you call it glass beads, Line 545) - is there a typically particle size of the sand? It would be useful to have this information to judge the type/amount of forces being applied to the biofilm to separate the robust and fragile fractions.

The methods were updated with relevant information.

Line 148 and Figure 2: you talk about the comparison to a non-gfp control - but you don't show this control. It needs to be added to the figures and/or added as a supplemental figure.

Line 156 and Figure 2: again you talk about the control level but don't show it, it needs to be added somewhere to show that the differences shown are true

Control histogram (obtained for non-labelled strain), was added to the bottom graphs (48h) as an extra layer colored red. Figure legend was updated accordingly.

Line 191 and Figure 3: you talk about background noise, but you do not show the background on the graphs. Was the non-fluorescent background pellicle only analysed once? Or was a separate pellicle analysed at each time point? Does the background levels change with time (ie does any autofluorescence appear/increase?) Could it be related to the presence of more EPS itself in the robust fraction?

*Control histograms were added. Unfortunately, due to practical challenges of time-series experiment, we limited the non-labelled control sampling to single datapoint – 48h-old pellicle, assuming that the highest-likelihood for background fluorescence for the mature pellicle. As control fluorescence values for the entire pellicle (robust and fragile fractions pulled) are rather low, they cannot explain rather substantial difference between robust and fragile fractions observed after 48h (especially for *tasA* expression).*

Figure 2: The peak shifts in the *eps* readouts at 20hour are unusual - the background non gfp control is also needed to be shown. Also if the same shift occurred in the *tapA* readout (which the ON population looks like it has compared to 16hour) then there may be a small OFF peak in this sample

*Control was added into the figure. The P_{eps} peak shifts between 16-20h, have been consistently observed across 3 biological replicates. Although such may seem unusual, they indicate increase intensity of intracellular GFP, which may result from a) accumulation of GFP in cells, because cell division slows down while GFP is still synthesized and not degraded or b) increase production rate of GFP due to higher activity of *eps* promoter. Looking at the pellicle growth data (Fig 1D), we favor the second scenario, since 16-20h represent the timeframe where pellicle biomass increases rapidly. We agree that small number of *tasA*-OFF cells may still be present at 20h. We commented on this aspect in the discussion in the revised manuscript version.*

Figure 4B: Wt only sample: why is the T=0 liquid sample CFU value so high only in this one sample?

We appreciate this note. After careful analysis of raw experimental data and comparison to the same experiment performed at a different day, we are convinced that unusually high starting CFU resulted from 'human error' in dilution series by a factor of 100x. Therefore, we decided to include data from additional experiments at different days, the figure was updated with accurate T=0 data. We took a liberty in such correction since data in 4B derive from destructive sampling, where each data point represents an independent pellicle.

Figure 5: There is a lot of variation between the results with different starting ratios - whilst there are clear trends, there are some obvious outliers (eg 0.84 ratio sample in *tasA* vs WT). Can this be explained or is it simply an experimental anomaly?

We believe that the variation between different ratios is a combination of biological variation and data analysis approach, where, we did not group the data, but showed each individual biofilm with its specific mutant vs WT ratio and corresponding absolute numbers of each strain. Statistical

analysis (Pearson's correlation coefficient) helps to understand significant relationships between mutant% and biofilm performance, as well as random effects.

Minor text changes:

Line 43: 'raise' should be 'raises'

Line 70: 'member of probiotic' should be 'member of a probiotic'

Line 73: 'Formation of pellicle biofilm' should either be 'Formation of a pellicle biofilms' or 'Formation of pellicle biofilms'

Line 74: 'towards liquid-air' should be 'towards the liquid-air'

Line 74: 'biofilm, in' should be 'biofilms, in'

Line 77: 'of extracellular' should be 'of the extracellular'

Line 88: 'variable and local' should be 'variable local'

Line 105: 'that protein' should be 'that the protein'

Line 119: 'magnifications' should be 'magnification'

Line 119: 'persisted up' should be 'persisted for up'

Line 121: can you add some timings to 'mature'?

Line 128: 'Most' should be 'At most'

Line 133: add hours after 24-48

Line 134: 'when number' should be 'when the number'

Line 134: 'and cells of fragile' should be 'and cell in the fragile'

Line 136: 'from emergence' should be 'from the emergence'

Line 152: 'of ON-' should be 'of the ON-'

Line 157: 'towards higher' should be 'towards a higher'

Line 158: 'with OFF-' should be 'with the OFF-'

Line 160: 'where OFF-' should be 'where the OFF-'

Line 161: 'points, heterogeneity' should be 'points, the heterogeneity'

Line 185: 'Pellicle with' should be 'pellicles of ages'

Line 192: 'towards low' should be 'towards a low'

Line 195: 'difference' should be 'differences'

Line 196: 'fragile' should be 'robust' based on the data?

Line 200: 'suggest' should be 'suggests'

Line 202: 'to increasing' should be 'to the increasing'

Line 203: 'seem' should be 'seems'

All the above were corrected

Line 207: you say observation of increased abundance of TasA - but you have not shown increased protein levels. You saw an increased number of delta-tasA cells?

Corrected to 'increased abundance of Δ tasA mutant'

Line 208: remove 'out'

Line 215: 'in both the' - you only say one sample, need to add in the pellicle as well?

Line 218: 'loosing' should be 'losing'

Line 220: 'that' should be 'which'

Lines 222-224: an extra line break here? This sentence is orphaned, and should be a part of the following paragraph?

Line 239: 'suggest' should be 'suggests'

Line 250: 'up' should be 'out'

Line 298: 'how extracellular' should be 'how the extracellular'

All above corrected.

Line 324: More info is needed here regarding vortexed - what settings were used? The force/speed of the vortex will have a big impact on the shear force applied

Information has been included in the materials and methods.

Line 349: Was the flow cytometry done immediately after sample collection or were the samples fixed and stored? Could this be the cause of the unusual changes in peak intensity at certain time points?

FC was done immediately after sample collection (methods updated), therefore the sample storage/fixing effects did not affect the result.

Line 381, 382 and 385, 556: 'access' or 'accessed' should be 'assess' or 'assessed'

Corrected

Line 549: you talk about blue and yellow bars, but the figure is in greyscale

Corrected

Table 1: There are strains used in the supplemental figures that are not included in the strains table, (eg the WT gfp strain), these should be added for completeness

The table containing the strains has been updated with the previously published strains. Thank you for catching this.

Figure 1C: In the text you talk about the different populations in terms of %, but the yaxis here is as a fraction - it would be better to be consistent

Corrected

Figure 2: The third repeat isn't visible in most of the graphs - could the colouring be changed to ensure all three repeats are visible?

We have attempted several coloring schemes; however, we find it extremely difficult to improve separation between the replicates, mainly because they often overlap each other. As an alternative, we could pick one representative replicate (common practice in presenting FC data), however we would prefer to show a 'full picture' and biological variability. Another solution would be to split the replicates into individual images, but at the cost of figure clarity. Therefore, we would prefer to maintain current data representation scheme. As results of statistical analysis are part of Fig. 2, they provide additional information on the level of overlap/variability between individual replicates.

Figure 4B: the volume that the pellicle was resuspended in should be given in the legend and/or the pellicle axes labelled simply as CFU/pellicle

Legend was updated

Figure 5: It might be helpful to also include a curve of 'total cell number' to see how the total

number of cells varies in each fraction of the biofilm with the different starting ratios, this may help with the anomalous samples like the 0.84 ratio in *tasA* vs WT

We agree with the above comment – total cell number data was included in the updated Fig. 5

Fig S1 Legend: 'accessed' should be 'assessed'

Corrected

Fig S1: were all the images taken at the same depth/height in the pellicle at all timepoints?

Images were taken thorough the entire depth of the pellicle and the Z-stack with highest GFP intensity was selected for figure representation (figure legend was updated accordingly).

Fig S3: A scale bar is needed for these images. Why has this figure been false-coloured blue and yellow when all others are in green/red? Yellow and blue is preferential for colour blindness, but it would be best to be consistent.

The scale bar has been emphasized. We have now adjusted the colors in the remaining figures to match the color-blind-friendly scheme.

July 13, 2020

Prof. Ákos T Kovács
Technical University of Denmark
Department of Biotechnology and Biomedicine
Søltofts Plads 221
Kgs Lyngby 2800
Denmark

Re: mSystems00425-20R1 (Privatization of biofilm matrix in structurally heterogeneous biofilms)

Dear Prof. Ákos T Kovács:

Your manuscript has been accepted, and I am forwarding it to the ASM Journals Department for publication. For your reference, ASM Journals' address is given below. Before it can be scheduled for publication, your manuscript will be checked by the mSystems senior production editor, Ellie Ghatineh, to make sure that all elements meet the technical requirements for publication. She will contact you if anything needs to be revised before copyediting and production can begin. Otherwise, you will be notified when your proofs are ready to be viewed.

Sincerely,

Mark Mandel
Editor, mSystems

Journals Department
Dataset 1: Accept

Fig S2: Accept

Fig S4: Accept

Fig S1: Accept

Supplemental Material: Accept

Fig S3: Accept